# Dendritic normalisation improves learning in sparsely connected artificial neural networks

**Alex D. Bird** [1,2,3]*, **Peter Jedlicka** [2,3], **Hermann Cuntz** [1,2]

**1** Ernst Strüngmann Institute for Neuroscience (ESI) in co-operation with Max Planck Society, Frankfurt, Germany, **2** Frankfurt Institute for Advanced Studies (FIAS), Frankfurt, Germany, **3** ICAR3R-Interdisciplinary Centre for 3Rs in Animal Research, Faculty of Medicine, Justus Liebig University Giessen, Giessen, Germany

* bird@fias.uni-frankfurt.de

**Data Availability Statement:** All code is freely available for download (see S1 Code and the Github repository 'Dendritic normalisation', https://github.com/synapsesanddendrites/Dendritic-

## Abstract

Artificial neural networks, taking inspiration from biological neurons, have become an invaluable tool for machine learning applications. Recent studies have developed techniques to effectively tune the connectivity of sparsely-connected artificial neural networks, which have the potential to be more computationally efficient than their fully-connected counterparts and more closely resemble the architectures of biological systems. We here present a normalisation, based on the biophysical behaviour of neuronal dendrites receiving distributed synaptic inputs, that divides the weight of an artificial neuron's afferent contacts by their number. We apply this dendritic normalisation to various sparsely-connected feedforward network architectures, as well as simple recurrent and self-organised networks with spatially extended units. The learning performance is significantly increased, providing an improvement over other widely-used normalisations in sparse networks. The results are two-fold, being both a practical advance in machine learning and an insight into how the structure of neuronal dendritic arbours may contribute to computation.

## Author summary

Neurons receive contacts from other cells on extensively branched processes known as dendrites. When a contact is formed, activity in one cell is communicated to another by altering the conductance of the receiving cell's membrane and allowing an ionic current to flow. A neuron with longer dendrites is intrinsically less excitable as these currents can more easily dissipate both across the larger cell membrane and along the dendrites themselves. We have recently shown that, in real neurons, this effect is precisely cancelled by the increased number of contacts allowed by longer dendrites. This in turn implies that the ability of a single synapse to influence a neuron is likely to be inversely proportional to the total number of synapses that that neuron receives. Here we study the computational implications of this effect using the well-established framework of artificial neural networks.

Sparsely-connected artificial neural networks adapt their connectivity to solve defined computational tasks such as classifying inputs and are at the forefront of modern machine learning. We apply the normalisation implied by dendritic structure to such networks:

normalisation). The MNIST and MNIST-Fashion datasets are included with the Github code. These can also be downloaded from various places, including at the time of writing, yann.lecun.com/exdb/mnist/ and github.com/zalandoresearch/fashion. Code for Figs 1 to 4 is written in Python 3.6. The networks in Figs 1, 2 and 4 are coded using the standard Numpy package, and the networks in Fig 3 make use of Keras with a TensorFlow backend (keras.io). The application of dendritic normalisation in Keras with TensorFlow allows for immediate inclusion in Keras-based deep learning models. The normalisation requires a custom layer, constraint, and optimiser. Fig 5 uses code written in Matlab 2020b, using the freely available Trees Toolbox package.

**Funding:** We acknowledge funding through BMBF grants 01GQ1406 (Bernstein Award 2013) to HC and 031L0229 to PJ. The funders had no role in study design, data collection and analysis, decision to publish, or preparation of the manuscript.

**Competing interests:** The authors have declared that no competing interests exist.

artificial neurons receiving more contacts require larger dendrites and so each individual contact will have proportionately less influence. Our normalisation allows networks to learn desired tasks faster and more consistently. Our result is both a practical advance in machine learning and a previously unappreciated way in which intrinsic properties of neurons may contribute to their computational function.

## Introduction

Artificial neural networks have had a huge impact over the last couple of decades, enabling substantial advances in machine learning for fields such as image recognition [1], language translation [2], and medical diagnosis [3]. The inspiration for these tools comes from biological neuronal networks, where learning arises from changes in the strength of synaptic connections between neurons through a number of different plasticity mechanisms [4–7]. The development of artificial neural networks away from the limitations of biology has meant that state-of-the-art artificial intelligence algorithms differ fundamentally from the biological function of the brain. For example global backpropagation algorithms have access to information that may be unavailable to real synaptic connections [8] (but see also [9]). Nevertheless, a number of biophysical principles have been successfully reintroduced, using salient features of real neuronal networks to make advances in the field of artificial neural networks [10–17]. Here we show how the dendritic morphology of a neuron, which influences both its connectivity and excitability, produces an afferent weight normalisation that improves learning in such networks.

Real neurons receive synaptic contacts across an extensively branched dendritic tree. Dendrites are leaky core conductors, where afferent currents propagate along dendritic cables whilst leaking across the cell membrane [18]. Larger dendrites increase the number of potential connections a cell can receive, meaning that more afferent currents can contribute to depolarisation [19, 20]. Conversely, larger cells typically have lower input resistances, due to the increased spatial extent and membrane surface area, meaning that larger synaptic currents are necessary to induce the same voltage response and so bring a neuron to threshold [21, 22]. It has recently been shown theoretically by Cuntz et al (2019) [23] that these two phenomena cancel each other exactly: the excitability of neurons receiving distributed excitatory synaptic inputs is largely invariant to changes in size and morphology. In addition, neurons possess several compensatory mechanisms to help maintain firing-rate homeostasis through both synaptic plasticity regulating inputs [24, 25] and changes in membrane conductance regulating responses [26, 27]. These results imply a consistent biophysical mechanism that contributes to stability in neuronal activity despite changes in scale and connectivity. We find that this mechanism is general and demonstrate it for artificial neural networks trained using backpropagation. The goal here is two-fold: firstly we produce results that outperform the current state-of-the-art for sparsely connected networks and secondly we demonstrate that learning is improved by dendrites in the ideal case where all gradient information is available to every synapse, as is the case with the traditional backpropagation algorithm.

Changing connectivity has traditionally not played much of a role in feedforward artificial neural networks, which typically used fully-connected layers where each neuron can receive input from all cells in the preceding layer. Sparsely-connected layers have, however, long been used as alternatives in networks with a variety of different architectures [10]. Sparse connectivity more closely resembles the structure of real neuronal networks and a number of recent studies have demonstrated that larger, but sparsely-connected, layers can be more efficient

than smaller fully-connected layers both in terms of total parameter numbers and training times [28–30]. The advantage in efficiency comes from the ability to entirely neglect synaptic connections that do not meaningfully contribute to the function of the network. The dendritic normalisation analysed in this paper has particular application here as it implies that relative synaptic plasticity will depend on the number of connections that a neuron receives, not necessarily their strengths. Sparse networks are also less likely to be overfitted to their training data as sparse representations of inputs are forced to focus on essential features of the signal instead of less-informative noise.

To produce an appropriate sparse connectivity a number of regularisation techniques have been suggested; $L^1$- and $L^0$-regularisations [28, 31] both penalise (the latter more explicitly) the number of connections between neurons during training. Mocanu et al (2018) [29], building on previous work [32–34], have recently introduced an evolutionary algorithm to reshape sparse connectivity, with weak connections being successively excised and randomly replaced. This procedure applies to both feedforward and recurrent artificial networks, causing feedforward networks to develop connectivities based on the properties of their inputs and recurrent networks to develop small-world and scale-free topologies similar to biological neuronal circuits [35]. Such networks have comparable performance to fully-connected layers, despite having many fewer parameters to optimise.

Normalisation is another feature that has previously been shown to enhance learning in neural networks. In particular Ioffe & Szegedy (2015) [36] introduced batch normalisation, where the inputs over a given set of training data are normalised, and Salimans & Kingma (2016) [37] introduced $L^2$-normalisation, where afferent synaptic weights are normalised by their total magnitude. The latter is reminiscent of heterosynaptic plasticity, where afferent synapses across a neuron depress in response to potentiation at one contact in order to maintain homeostasis [25, 38, 39]. Both techniques have been applied in fully-connected networks and both work to keep neuronal activities in the region where they are most sensitive to changes in inputs. Interestingly, existing studies of sparse networks do not typically include any normalisation. The normalisation that arises from the relationship between a real neuron's morphology and connectivity provides a particularly powerful, and biologically realistic, way to normalise sparse inputs. Dividing the magnitude of individual synaptic weights by their number distributes activity across neurons whilst keeping each cell sensitive to changes in inputs; neurons therefore encode signals from the proportion of presynaptic partners that are active, providing a simple and broadly applicable technique to ensure faster convergence to optimal solutions.

## Results

### Dendritic normalisation is an intrinsic property of spatially extended neurons

Cuntz et al (2019) [23] have shown that neuronal excitability in response to distributed synaptic input is invariant of size. This invariance is exact for a homogeneous passive cable and holds approximately for realistic heterogeneities in dendritic diameter, topology, input dynamics, and active properties. We here extend the major result of that study to demonstrate explicitly how the influence of individual synapses, and of local changes in synaptic strength, depend on neuronal size and hence afferent connectivity.

Given a closed dendritic cable of physical length $l$ with constant radius $r$, axial resistivity $r_a$, and membrane conductivity $g_l$, the electronic length can be written as $L = l/\lambda$ for length constant $\lambda = \sqrt{\frac{r}{2r_a g_l}}$. The steady-state voltage at the root in response to a constant current influx of

unit magnitude at electrotonic distance $X$ is given by the transfer resistance $R_L(X)$

$$R_L(X) = \frac{1}{G_\infty}\left[\frac{\cosh(L-X)}{\sinh(L)}\right] \tag{1}$$

where $G_\infty = \frac{\pi r^2}{\lambda r_a}$ is the input conductance at the sealed end of a semi-infinite dendrite with the above properties. Given that afferent currents are typically integrated at the soma, a change in magnitude $\Delta_{syn}$ at a synapse at location $X$ will result in a change in somatic voltage in response to synaptic activation of $\Delta_v = R_L(X)\Delta_{syn}$. If inputs are distributed uniformly randomly along the dendrite, then the mean (over all possible synaptic locations) $\mu_{\Delta_v}$ and variance $\sigma^2_{\Delta_v}$ of the change in somatic voltage response to a perturbation of local synaptic strength are

$$\mu_{\Delta_v} = \frac{\Delta_{syn}}{L\,G_\infty} \qquad , \qquad \sigma^2_{\Delta_v} = \frac{\Delta^2_{syn}}{L^2\,G^2_\infty}\left(L^2\coth(L) + L\,\mathrm{cosech}^2(L) - 1\right) \tag{2}$$

There is thus an inverse relationship between neuronal size and the expected effect of a given change in local synaptic weight on the somatic voltage, and hence spiking probability.

Dendrites receive synapses. In many cell types, dendrites that are no longer innervated retract [42] and the statistical properties of neurite arbours imply a proportional relationship between dendrite length and potential afferent connectivity, both for general axonal inputs [43] and between individual pairs of neurons [20] (Fig 1A). The inverse relationship between dendritic length and spiking probability therefore feeds into an inverse relationship between the number of synaptic contacts a neuron receives, $n$, and the magnitude of any change in neuron excitability resulting from local synaptic plasticity.

$$\mu_{\Delta_v} \propto \frac{1}{L} \propto \frac{1}{n} \tag{3}$$

This is the basis of dendritic normalisation. Similar results hold if the size of the soma is increased (Eq 11 and Fig 1B, top panel, dashed lines), or synaptic transients are considered (Eqs 14 and 18 and Fig 1B, middle and lower panels).

## Dendritic normalisation and stochastic gradient descent

Let $\mathbf{w}_i$ be the input weight vector to a given neuron $i$. Then the dendritic normalisation above can be written as

$$\mathbf{w}_i = \frac{s}{\|\mathbf{v}_i\|_0}\mathbf{v}_i \tag{4}$$

where $\mathbf{v}_i$ is an unnormalised weight vector of the same size as $\mathbf{w_i}$, $\|\mathbf{v_i}\|_0$ is the $L^0$-norm of $\mathbf{v}_i$ (the number of non-zero elements), and $s$ is a scalar that determines the magnitude of the vector $\mathbf{w}_i$. The parametrisation here differs from that introduced by Salimans & Kingma (2016) [37] for fully-connected networks with Euclidean normalisation in two fundamental ways. Firstly, the magnitude parameter $s$ is the same across all neurons as it reflects a conserved relationship between connectivity and excitability. If a network were to include distinct classes of artificial neurons with distinct synaptic integration properties, different values of $s$ may be appropriate for each class, but should not differ between neurons of the same class. Secondly, the $L^0$-norm $\|\mathbf{v_i}\|_0$ is distinct from the Euclidean $L^2$-norm in that it is almost surely constant with respect to $\mathbf{v}_i$: Connections are not created or destroyed by stochastic gradient descent.

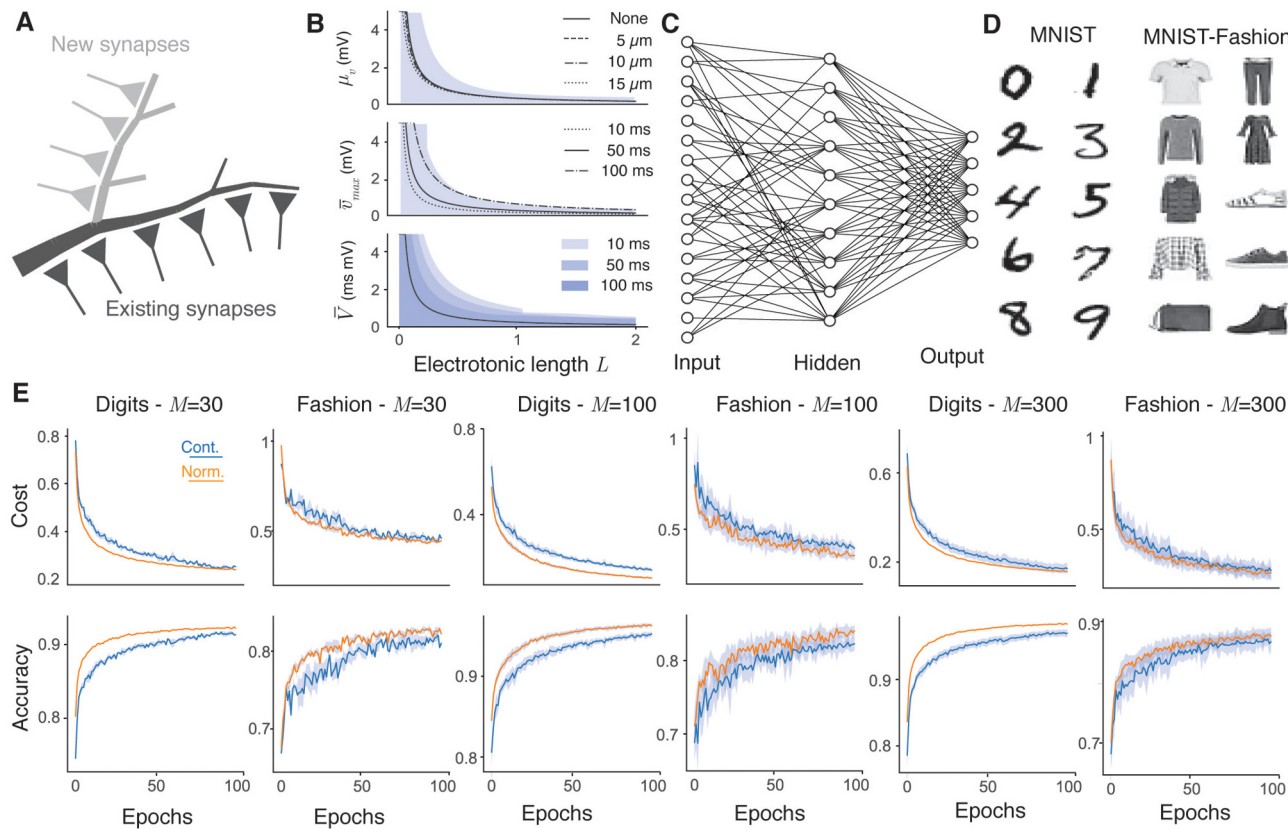

**Fig 1. Dendritic normalisation improves learning in sparse artificial neural networks. A**, Schematic of dendritic normalisation. A neuron receives inputs across its dendritic tree (dark grey). In order to receive new inputs, the dendritic tree must expand (light grey), lowering the intrinsic excitability of the cell through increased membrane leak and spatial extent. **B**, Expected impact of changing local synaptic weight on somatic voltage as a function of dendrite length and hence potential connectivity. Top: Steady state transfer resistance (Eq 10) for somata of radii 0, 5, 10, and 15 $\mu m$. Shaded area shows one standard deviation around the mean in the 0 $\mu m$ case (Eq 11). Middle: Maximum voltage response to synaptic currents with decay timescales 10, 50, and 100 ms (Eqs 14 and 16). Shaded area shows one standard deviation around the mean in the 100 ms case (Eq 15). Bottom: Total voltage response to synaptic currents with the above timescales (all averages lie on the solid line, Eq 17). Shaded areas show one standard deviation around the mean in each case (Eq 18). Intrinsic dendrite properties are radius $r = 1\ \mu m$, membrane conductivity $g_l = 5 \times 10^{-5}$ S/cm$^2$, axial resistivity $r_a = 100\ \Omega$cm, and specific capacitance $c = 1\ \mu$F/cm$^2$ in all cases and $\Delta_{syn} = 1$ mA. **C**, Schematic of a sparsely-connected artificial neural network. Input units (left) correspond to pixels from the input. Hidden units (centre) receive connections from some, but not necessarily all, input units. Output units (right) produce a classification probability. **D**, Example $28 \times 28$ pixel greyscale images from the MNIST [40] (left) and MNIST-Fashion [41] (right) datasets. The MNIST images are handwritten digits from 0 to 9 and the MNIST-Fashion images have ten classes, respectively: T-shirt/top, trousers, pullover, dress, coat, sandal, shirt, sneaker, bag, and ankle boot. **E**, Learning improvement with dendritic normalisation (orange) compared to the unnormalised case (blue). Top row: Log-likelihood cost on training data. Bottom row: Classification accuracy on test data. From left to right: digits with $M = 30$ hidden neurons, fashion with $M = 30$, digits with $M = 100$, fashion with $M = 100$, digits with $M = 300$, fashion with $M = 300$. Solid lines show the mean over 10 trials and shaded areas the mean ± one standard deviation. SET hyperparameters are $\varepsilon = 0.2$ and $\zeta = 0.15$.

The gradients of a general cost function $C$ with respect to $\mathbf{v}_i$ and $s$ can be written as

$$\nabla_{\mathbf{v}_i} C = \frac{s}{\|\mathbf{v}_i\|_0} \nabla_{\mathbf{w}_i} C \quad , \quad \frac{\partial C}{\partial s} = \sum_{\mathbf{w}_i} \frac{1}{\|\mathbf{v}_i\|_0} \nabla_{\mathbf{w}_i} C \cdot \mathbf{v}_i \tag{5}$$

where $\nabla_{\mathbf{w}_i} C$ is the usual gradient of $C$ with respect to the full weight vector $\mathbf{w}_i$ and the sum in the second equation is over all weight vectors in the network. Note that $\|\mathbf{v}_i\|_0$ is almost everywhere constant during gradient descent as it is the number of non-zero elements of $v_i$ (although this may change between epochs under the evolutionary connectivity algorithm). Examples of $C$ are given in the Methods (Eqs 22 and 24). An interesting consequence of these

equations is that neurons with more afferent connections will have smaller weight updates, and so be more stable, than those with fewer afferent connections.

## Dendritic normalisation improves learning in sparsely-connected feedforward neural networks

We consider the performance of sparse neural networks with and without dendritic normalisation on the MNIST and MNIST-Fashion datasets (Fig 1C and 1D). Connections between the input and hidden layers are established sparsely (Fig 1C) using the sparse evolutionary training (SET) algorithm introduced by Mocanu et al [29]. Briefly, connections are initiated uniformly randomly with probability $\varepsilon$ to form an Erdős-Rényi random graph [44]. Each neuron will therefore receive a variable number of afferent contacts. After each training epoch, a fraction $\zeta$ of the weakest contacts in the entire layer are excised and an equal number of new random connections are formed between different neurons. This means that the number of connections received by each neuron will typically change between epochs. New connection weights are distributed normally with mean 0 and standard deviation 1.

For sparse networks, trained using stochastic gradient descent and SET, with one hidden layer consisting of 30, 100, and 300 neurons, with connection probability $\varepsilon = 0.2$ and SET excision rate $\zeta = 0.15$, the normalised network consistently learns faster than the unnormalised control network (orange lines against blue lines). This result holds across both the cost on the training sets and the accuracy on the test sets for both datasets and across all network sizes, indicating a robust improvement in learning performance. In addition, the variability between different independent training regimes (shaded areas in Fig 1E show mean plus or minus one standard deviation over 10 independently initiated training regimes) is reduced for these experiments.

## Evolution of connectivity

It is possible to visualise the connection structure that results from training the control and normalised networks on the MNIST data (Fig 2). Fig 2A shows how the number of efferent connections from each input neuron (organised as pixels) changes with the number of training epochs. Initially, the connections are randomly distributed and have no spatial structure, but the SET algorithm gradually imposes a heavier weighting on the central input neurons as training progresses. This feature was shown before by Mocanu et al (2018) [29] as central pixels are likely to be more informative over the relevant datasets. Comparing the control (left, blue) and normalised (right, orange) networks, it is interesting to note that the bias towards central pixels is less strong in the normalised case: Input neurons over a relatively broad area are strongly connected to the hidden layer.

Postsynaptically, the number of contacts received by each hidden neuron is less variable in the normalised case (Fig 2B, left column), the weights of these contacts are typically smaller in absolute value and less dispersed (Fig 2B, central column); the resultant weighted inputs over the test data to hidden neurons are therefore more consistent (Fig 2B, right column). The normalisation appears to make better use of neural resources by distributing connections and weights more evenly across the available cells, whilst keeping expected inputs closer to 0, the steepest part of the activation function, where responses are most sensitive to inputs, during early training. In addition, the smaller connection weights suggest that normalised networks may be even more robust to overfitting than the equivalent unnormalised sparse network [31]. This is supported by the increased improvement in the case of more complex networks, both in terms of more hidden units and more layers, as well as the greater improvement in evaluation accuracy compared to training cost (Figs 1 and 3).

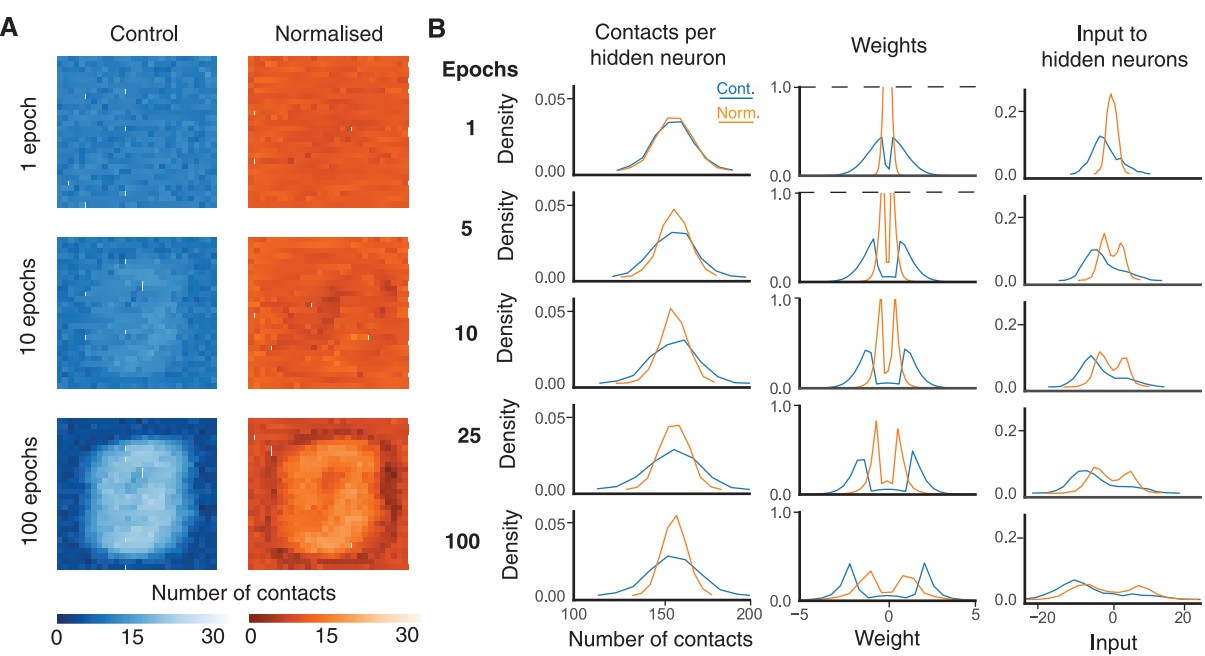

**Fig 2. Evolution of synaptic weights. A**, Number of efferent contacts from each input neuron (pixel) to neurons in the hidden layer as the weights evolve. The left panels (blue) show the unnormalised case and the right (orange) the normalised case. **B**, Afferent contacts for the unnormalised (blue) and normalised (orange) cases. From left to right: Distribution of the number of afferent contacts arriving at each hidden neuron, weights, and mean weighted input to each hiddden neuron over the test set. All panels show the average over 10 trials on the original MNIST dataset with hyperparameters $M = 100$, $\varepsilon = 0.2$, and $\zeta = 0.15$. Dashed lines show where the vertical axis has been truncated to preserve the scale.

## Improved training in deeper feedforward networks

The improvement in learning performance generalises to deeper networks with multiple hidden layers (Fig 3A) and different levels of sparsity (S1 Fig), as well as convolutional networks with a sparsely connected layer following the max pooling layer (Fig 3C). In all cases, final classification performance and training speed are improved by dendritic normalisation (orange versus blue lines in Fig 3B and 3D for the MNIST-Fashion dataset). Interestingly, the reduction in trial-to-trial variability seen in single-layer networks (Fig 1) does not occur here. Dendritic normalisation is therefore applicable, and beneficial, as a universal technique for sparse layers in deep networks. The improvement in performance is not limited to artificial neurons with a sigmoid activation function. When neurons instead have non-saturating threshold linear activations the dendritic normalisation again improves learning (Fig 3E).

## Comparison with other normalisations

Dendritic normalisation is just one of many mechanisms to promote and stabilise learning and activity in real neurons. Real neurons are likely to experience not only passive dendritic normalisation but also active modifications of synaptic weights under heterosynaptic plasticity [25, 38]. We study this in artificial neurons by combining the $L^0$-normalisation introduced here with the $L^2$-normalisation of Salimans & Kingma (2016) [37]. Here, inputs to a cell are normalised by both number and magnitude

$$\mathbf{w}_i = \frac{g_i}{\|\mathbf{v}_i\|_0 \|\mathbf{v}_i\|_2} \mathbf{v}_i \tag{6}$$

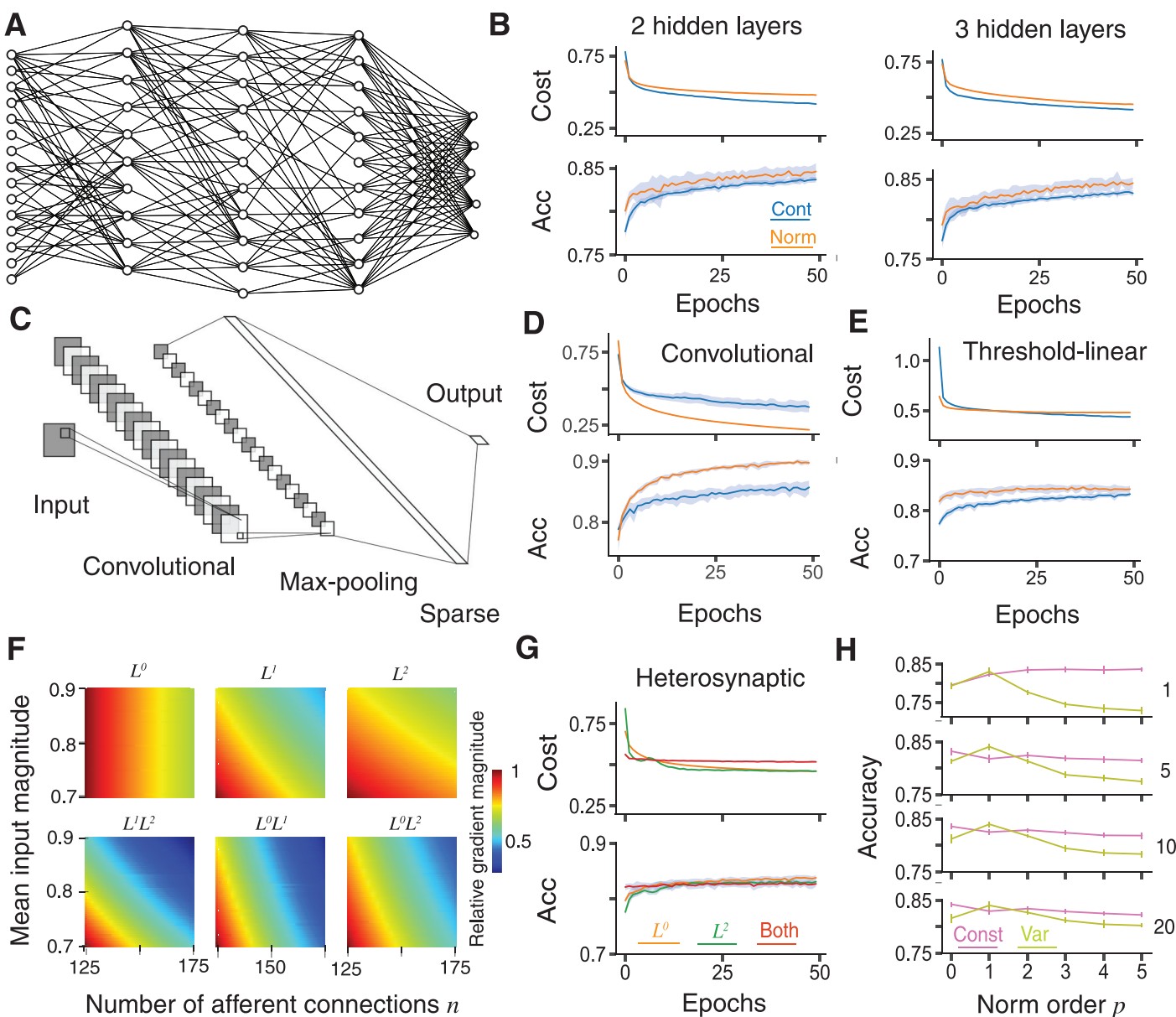

**Fig 3. Improved training in deeper networks and comparison with other norms. A**, Schematic of a sparsely connected network with 3 hidden layers. The output layer is fully connected to the final hidden layer, but all other connections are sparse. **B**, Learning improvement with dendritic normalisation (orange) compared to the unnormalised control case (blue) for networks with 2 (top) and 3 (bottom, see panel **A**) sparsely-connected hidden layers, each with $M = 100$ neurons. Top of each: Log-likelihood cost on training data. Bottom of each: Classification accuracy on test data. **C**, Schematic of a convolutional neural network [46] with 20 5 × 5 features and 2 × 2 maxpooling, followed by a sparsely connected layer with $M = 100$ neurons. **D**, Improved learning in the convolutional network described in **C** for an unnormalised (blue) and normalised (orange) sparsely-connected layer. Top: Log-likelihood cost on training data. Bottom: Classification accuracy on test data. **E**, Improved learning in a network with one hidden layer with $M = 100$ threshold-linear neurons for unnormalised (blue) and normalised (orange) sparsely-connected layers. Top: Log-likelihood cost on training data. Bottom: Classification accuracy on test data. **F**, Contribution of different norm orders to the learning gradients of neuron with different numbers of afferent connections and different mean absolute connection weights. Norms are (left to right and top to bottom): $L^0$ (dendritic normalisation), $L^1$, $L^2$ [37], joint $L^1$ and $L^2$, joint $L^0$ and $L^1$, and joint $L^0$ and $L^2$ (Eq 6). Values are scaled linearly to have the a maximum of 1 for each norm order. **G**, Comparison of dendritic (orange), heterosynaptic (green [37]), and joint (red, Eq 6) normalisations. Top: Log-likelihood cost on training data. Bottom: Classification accuracy on test data. **H**, Comparison of test accuracy under different orders of norm $p$ after (from top to bottom) 1, 5, 10, and 20 epochs. Pink shows constant (Eq 8) and olive variable (Eq 9) excitability. Solid lines show the mean over 20 trials and shaded areas and error bars the mean ± one standard deviation. All results are on the MNIST-Fashion dataset. Hyperparameters are $\varepsilon = 0.2$ and $\zeta = 0.15$.

where $\|\mathbf{v}_i\|_2 = \sqrt{\sum_{j=1}^{n_i} |v_{i,j}|^2}$ is the $L^2$-norm of $\mathbf{v}_i$. The uniform excitability parameter $s$ is now contained in the individual magnitudes $g_i$ that differ for each cell. The gradients of the cost function $C$ with respect to $\mathbf{v}_i$ and $g_i$ can now be written as

$$\nabla_{\mathbf{v}_i} C = \frac{g_i}{\|\mathbf{v}_i\|_0 \|\mathbf{v}_i\|_2} \left( \nabla_{\mathbf{w}_i} C - \frac{\nabla_{g_i} C}{\|\mathbf{v}_i\|_2} \mathbf{v_i} \right) \quad , \quad \nabla_{g_i} C = \frac{g_i}{\|\mathbf{v}_i\|_0 \|\mathbf{v}_i\|_2} \nabla_{\mathbf{w}_i} C \cdot \mathbf{v_i} \quad (7)$$

With both normalisations applied, well-connected neurons will again tend to change afferent weights more slowly, as will cells with lower excitability $g_i$. In consequence, both cells receiving a few strong connections and those receiving many, individually less effective, connections will be more stable and relatively slow learning. The relative strength of these different plasticities are plotted as functions of both contact number and mean absolute weight in Fig 3F. Here we can see the relative gradient magnitudes that would be expected to arise from using different norms on neurons receiving sparse afferents with probability $\epsilon = 0.2$ and normally distributed weights from an input layer of size 784 (the standard MNIST size). All values are scaled linearly for comparison so that the largest magnitude for each norm is 1; in practice similar equalisations would be achieved by the automatic optimisation of the parameters $s$ and $g_i$ as described above. The smallest values for each norm can therefore differ. The top row shows the individual norms of different orders, whilst the bottom shows their combinations. Dendritic $L^0$-normalisation gives a gradient magnitude that is independent of mean connection weight, whereas the other norms depend on both features in combination. Combining two norms also typically leads to a larger relative difference in gradient magnitudes across the space of possible afferent connectivities.

The effects of dendritic (orange) and heterosynaptic (green) normalisations can be seen in Fig 3G. Here, a single layer of 100 hidden neurons is trained on the MNIST-Fashion data. Both normalisations individually have similar performance, with a slight advantage for the dendritic normalisation (orange): the mean test accuracy after 50 epochs is 83.75% compared to 82.66% for the heterosynaptic normalisation over 20 trials, and the accuracy is consistently higher. Interestingly, the joint normalisation given by Eq 6 (red lines in Fig 3G) reaches almost its highest level of accuracy after a single epoch and does not improve substantially thereafter. As neurons with many weak connections learn relatively slowly when both normalisations are in place (Fig 3F, lower right corners) it appears that good use is made of existing connectivity, but that less informative connections are not selectively weakened enough to be excised by the SET algorithm. When the $L^0$ norm alone is applied, the strength of the individual connections is irrelevant for the learning rate. The joint mechanism appears well suited to the sparse, but static, connectivity of the first epoch before SET has been applied whilst lacking the power of individual normalisations, either dendritic or heterosynaptic, to identify less necessary synapses. Whilst it is possible to exaggerate the biological relevance of learning through backpropagation, it is interesting to note that the dendritic $L^0$-normalisation is intrinsic to real neurons, whilst heterosynaptic-like $L^2$-normalisation can be regulated [25].

Biologically plausible learning rules often include normalisation of afferent weights to ensure stability and improve convergence. Both Oja's rule [6] and the unsupervised learning procedure introduced by Krotov & Hopfield (2019) [17], for example, ensure convergence to $L^p$-normalised afferent weights for $p \geq 2$. Whilst a direct comparison with such distinct learning rules is beyond the scope of this study, many such normalisations rely on the $L^p$ weight norm for some value of $p$ with either constant ($s$ in Eq 4) or variable ($g_i$ in Eq 6) excitability

ratios. For $p \geq 1$ the general gradients provided by such $L^p$-normalisations are

$$\nabla_{\mathbf{v}_i} C = \frac{s}{\|\mathbf{v}_i\|_p} \left( \nabla_{\mathbf{w}_i} C - \frac{\nabla_{\mathbf{w}_i} C \cdot \mathbf{v}_i}{\|\mathbf{v}_i\|_p^p} \mathbf{v}_{\mathbf{i}}^{p-1} \right) \qquad , \qquad \frac{\partial C}{\partial s} = \sum_{\mathbf{w}_i} \frac{1}{\|\mathbf{v}_i\|_p} \nabla_{\mathbf{w}_i} C \cdot \mathbf{v}_i \qquad (8)$$

for constant excitability and

$$\nabla_{\mathbf{v}_i} C = \frac{g_i}{\|\mathbf{v}_i\|_p} \nabla_{\mathbf{w}_i} C - \frac{g_i \nabla_{g_i} C}{\|\mathbf{v}_i\|_p^p} \mathbf{v}_{\mathbf{i}}^{p-1} \qquad , \qquad \nabla_{g_i} C = \frac{g_i}{\|\mathbf{v}_i\|_p} \nabla_{\mathbf{w}_i} C \cdot \mathbf{v}_{\mathbf{i}} \qquad (9)$$

for variable excitability. In all cases, higher $L^p$-norms imply slower learning; increasing the value of $p$ means that the norms $\|\mathbf{v}_i\|_p$ become increasingly insensitive to the smallest afferent weights.

Fig 3H shows the mean test accuracy for 100 hidden neurons on the MNIST-Fashion data after 1, 5, 10, and 20 training epochs as a function of the order $p$ of normalisation for $p = 0, 1, 2, 3, 4,$ and 5. The pink lines show the case of constant excitability (Eq 8) and the olive lines variable excitability (Eq 9). All normalisations show substantial improvement over the control case for the same task (Fig 1E) and performance is fairly similar across a broad range of parameters. The mechanism is likely to be similar in all cases by keeping neuronal activation within a useful range. The dendritic normalisation ($L^0$-norm in pink) and $L^1$-norm with variable excitability typically have the best accuracies after the first epoch. The mean accuracies ± one standard deviation for the constant excitability $L^0$- and variable excitability $L^1$-norms are $0.8360 \pm 0.0508$ and $0.8402 \pm 0.0055$ after 10 epochs and $0.8423 \pm 0.0040$ and $0.8402 \pm 0.0074$ after 20 epochs. The results are sufficiently consistent that the accuracies are significantly different in both cases, with $p$-values less than $10^{-5}$ under a Welch's T-test [45].

Interestingly, for all orders except $p = 1$, the constant excitability case has better performance than the situation with variable excitability and this gap increases with $p$. In the constant excitability case, a single parameter $s$ determines the postsynaptic response to afferent weights with a given norm, whereas in the variable case each neuron $i$ has its own response $g_i$ to normed inputs. It appears that maintaining a comparable excitability between neurons is superior in terms of learning to allowing this to vary between cells. This is an interesting observation as standard normalisations retain neuron-specific excitability, and this is assumed to be beneficial for their function [36, 37]. Constant excitability appears to act as a regularisation, finding the most appropriate single excitability to prevent overfitting (as can be seen in the differences between training cost and test accuracy in Fig 3G), and incorporates information about the gradients of all individual connection weights into a general tuning of the effective learning rate (Eq 5). A future study could identify precisely when these features are more beneficial than the greater flexibility inherent in neuron-specific excitability.

## Performance on standard benchmarks

As a final test in feedforward artificial neural networks, we show that dendritic normalisation can enhance the accuracy of artificial neural networks on common benchmark datasets. We compare dendritically normalised sparse networks to the published results of comparable sparse networks in Table 1, in each case replicating the published network size and hyperparameters. It should be noted that the sparse results quoted in the literature are often in the context of improving the performance of fully-connected neural networks with the same architecture despite having many fewer parameters. In most cases, applying dendritic normalisation improves upon the published performance. The one exception is the COIL-100 data where the

**Table 1. Table of performance for benchmark datasets compared to published results on sparse networks.** We replicate the published architecture in each case for a fair comparison: For the original MNIST dataset and CIFAR-10 datasets, Mocanu et al (2018) [29] used three sparsely-connected layers of 1000 neurons each and 4% of possible connections existing. Pieterse & Mocanu (2019) [30] used the same architecture for the COIL-100 dataset. For the Fashion-MNIST dataset, Pieterse & Mocanu (2019) [30] used three sparsely-connected layers of 200 neurons each, with 20% of possible connections existing.

| Dataset | Size | | | Accuracy | |
|---|---|---|---|---|---|
| | Training | Test | Classes | Control (Source) | Normalisation |
| MNIST [40] | 60, 000 | 10, 000 | 10 | 98.74% [29] | **99.63%** |
| Fashion-MNIST [41] | 60, 000 | 10, 000 | 10 | 89.01% [30] | **92.23%** |
| CIFAR-10 [67] | 50, 000 | 10, 000 | 10 | 74.84% [29] | **77.43%** |
| COIL-100 [68] | 5, 764 | 1, 436 | 100 | **98.68% [30]** | 98.47% |

training/test split is random; using slightly different data to Pieterse & Mocanu (2019) [30] could explain this difference.

## Sparse recurrent networks with backpropagation through time

A further application of sparsity lies in the field of recurrent artificial neural networks [7, 8], where intralayer connectivity allows information about previous inputs to persist and propagate. Such networks have found natural applications in speech recognition [47] and translation [2]. We consider the effect of dendritic normalisation in a network with sparse recurrent connectivity designed to perform binary addition (Fig 4A). The task is relatively simple, two input neurons apply a binary sequence of digits in the range from 0 to $2^{50}$ to a sparse recurrent layer of 50 units that must both remember the running total and incorporate the new information, but it nevertheless highlights the role of dendritic normalisation alongside this form of connectivity. As the dataset in this case is unlimited, epochs of length 1000 are used for the SET process. Here, both networks are well able to learn the task when outputs are rounded to the nearest integer (Fig 4B, bottom), but dendritic normalisation leads to a faster reduction in mean-squared error of the raw output (orange versus blue in (Fig 4B, top).

Interestingly, these results appear to arise from a notably different connectivity structure. After 100 epochs the control network develops a small number of highly-connected neurons and many less-connected neurons (Fig 4C, left, blue bars), whereas the dendritically normalised network favours many more neurons receiving inputs from all others and an input degree distribution that tails off towards lower afferent connectivity (Fig 4C, left, orange bars). Both networks develop relatively low average shortest path lengths, but the dendritic normalisation leads to especially short paths. The initial average shortest path length of the network is 1.7664 ± 0.0077 in both cases; the SET algorithm in the control case reduces this to 1.5278 ± 0.2155 and with dendritic normalisation it falls to 1.1458 ± 0.1830. In terms of weights, the dendritic normalisation again leads to relatively smaller magnitudes than the control case.

Combining a feedforward task with a recurrent one means that neurons must balance two competing streams of information. The above network can be adapted to include sparse feedforward connectivity (Fig 4D). In this case the spaces of all possible feedforward and recurrent connections are considered together so that weaker feedforward connections are effectively in competition with weaker recurrent connections to avoid excision by the SET algorithm. For the dendritic normalisation, different values of $s$ are used for the sets of feedforward and recurrent connections, potentially reflecting distinguishable synaptic types or input locations [48]. The sparsity of input makes the task slightly harder and both networks have lower accuracies than in the case of dense feedforward connectivity (Fig 4E, top). Again, the dendritic normalisation allows a faster reduction in mean-squared error of the raw output (orange versus blue

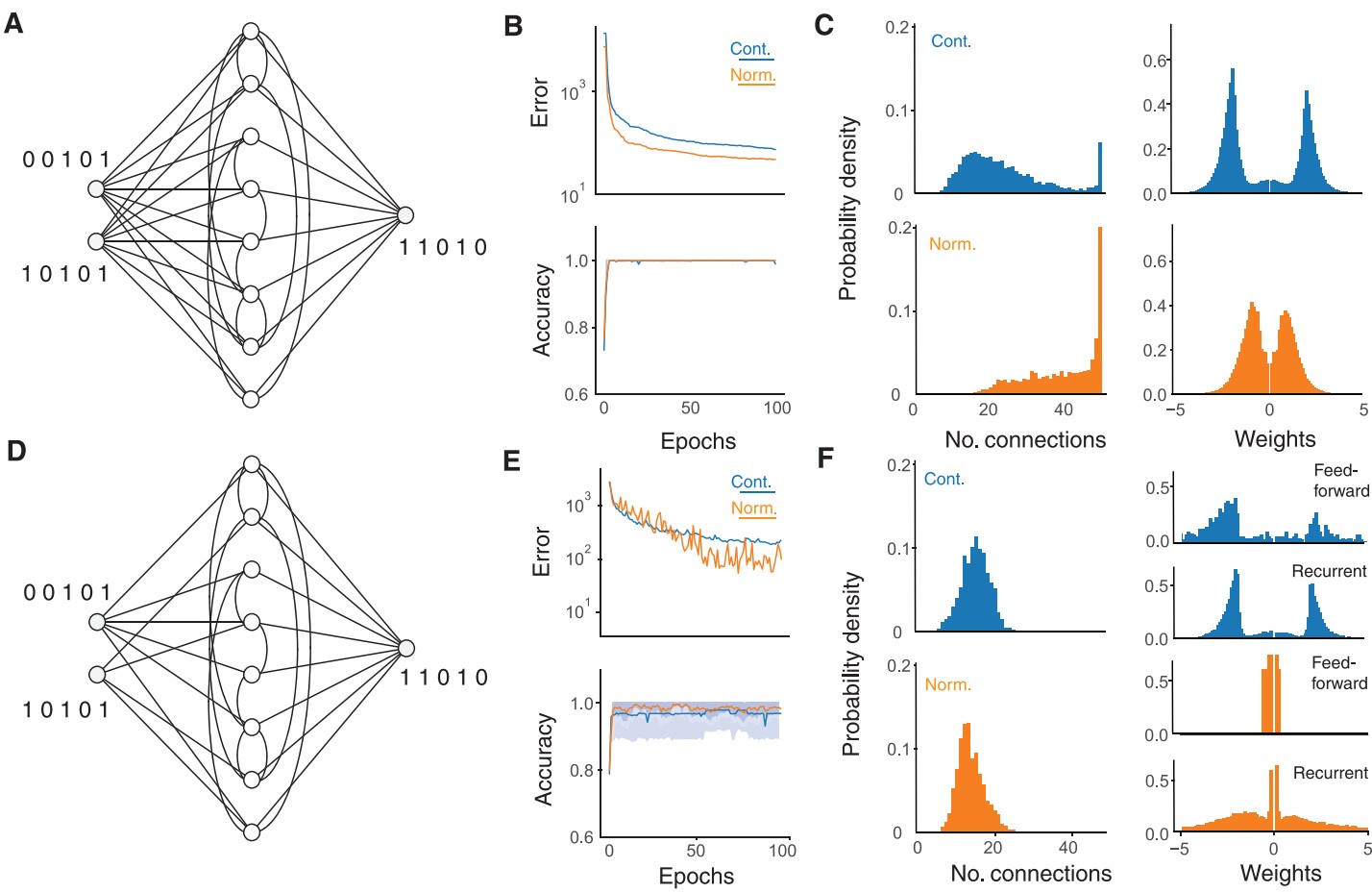

**Fig 4. Sparse recurrent networks with backpropagation through time. A**, Schematic of a network with dense feedforward and sparse recurrent connectivity. **B**, Learning improvement with dendritic normalisation (orange) compared to the unnormalised control case (blue) for the above network with $M = 50$ neurons adding binary numbers up to $2^{50}$. Top: Mean-square error cost. Bottom: Classification accuracy. Solid lines show average over 100 repetitions and shaded regions in the bottom graph show the mean ± on standard deviation (truncated to be below an accuracy of 1). **C**, Final distributions of afferent connectivity degrees (left) and weights (right) in each case after 100 epochs. **D**, Schematic of a network with sparse feedforward and recurrent connectivity. **E**, Learning improvement with dendritic normalisation (orange) compared to the unnormalised control case (blue) for the above network with $M = 50$ neurons adding binary numbers up to $2^{50}$. Top: Mean-square error cost. Bottom: Classification accuracy. Solid lines show average over 100 repetitions and shaded regions in the bottom graph show the mean ± on standard deviation (truncated to be below an accuracy of 1). **F**, Final distributions of recurrent afferent connectivity degrees (left), feedforward weights (right, top in each case), and recurrent weights (right, bottom in each case) after 100 epochs. Hyperparameters are $\varepsilon = 0.3$ and $\zeta = 0.15$.

in (Fig 4E, bottom). In this case, the connectivities do not develop the structures seen in the case when only the recurrent synapses are sparse and undergo SET (Fig 4F, left). The competition between the two streams of input eliminates the tendency for well-connected nodes to appear. It may well be beneficial for cells to structurally isolate inputs with different sources. Whilst in the control case, the sets of feedforward and recurrent weights develop similar magnitudes (Fig 4F, right top), the dendritic normalisation favours weak feedforward and strong recurrent connectivity (Fig 4F, right bottom). The weak feedforward connections are, however, much more widespread with 92.8 ± 4.4% of recurrent layer cells receiving connections from both input neurons in contrast to 17.0 ± 3.8% in the control case.

It should be noted that in contrast to the strictly feedforward case (Figs 1–3), dendritic normalisation leads to greater trial-to-trial variability in learning performance in the case of sparse feedforward and recurrent connectivity, despite typically producing better results (Fig 4E). It appears that the weaker feedforward connections are particularly prone to excision by the SET

protocol and that this destroys much of the information contained in their weights; a modification of the threshold for removal between the two sets of connections would be an interesting extension to address this issue.

## Self-organisation in networks of spatially extended spiking neurons

We have demonstrated that the normalisation properties of dendrites improve learning in sparsely-connected artificial neural networks with a variety of different architectures, both feedforward and recurrent. In all cases, supervised learning takes place using backpropagation with synaptic magnitudes and gradients constrained by explicit rules (Eqs 4 and 5). As a final example, we show that if spatially extended neurons are used it is not necessary to impose such external constraints; dendritic normalisation is an intrinsic property and naturally leads to better learning.

In Fig 5A we plot an example network of excitatory (green) and inhibitory (red) dendritic trees (see Methods for details). Axons are not treated explicitly and hence there are no spatial constraints on connectivity, but sparse current-based synapses are initially randomly distributed so that 30% of all possible excitatory-excitatory and 70% of both all possible excitatory-inhibitory and inhibitory-excitatory connections exist and impinge uniformly randomly on the dendritic trees of the postsynaptic cells. Dendrites are passive and the neurons are equipped with a spiking mechanism so that if the somatic voltage exceeds a certain threshold a spike is generated. Dendrites are then resized to match the number of their afferent contacts. As we now explicitly differentiate between excitatory and inhibitory contacts, we draw initial local synaptic absolute weights from a strictly positive gamma distribution (Eq 25) and assign the appropriate sign. The network learns a simple reservoir computing task inspired by that in [49]. Sequences of inputs are randomly presented to non-overlapping subpopulations of excitatory neurons and the recurrent excitatory-excitatory connectivity undergoes spike-timing dependent plasticity [5]. After a number of presentations of stimulus forming

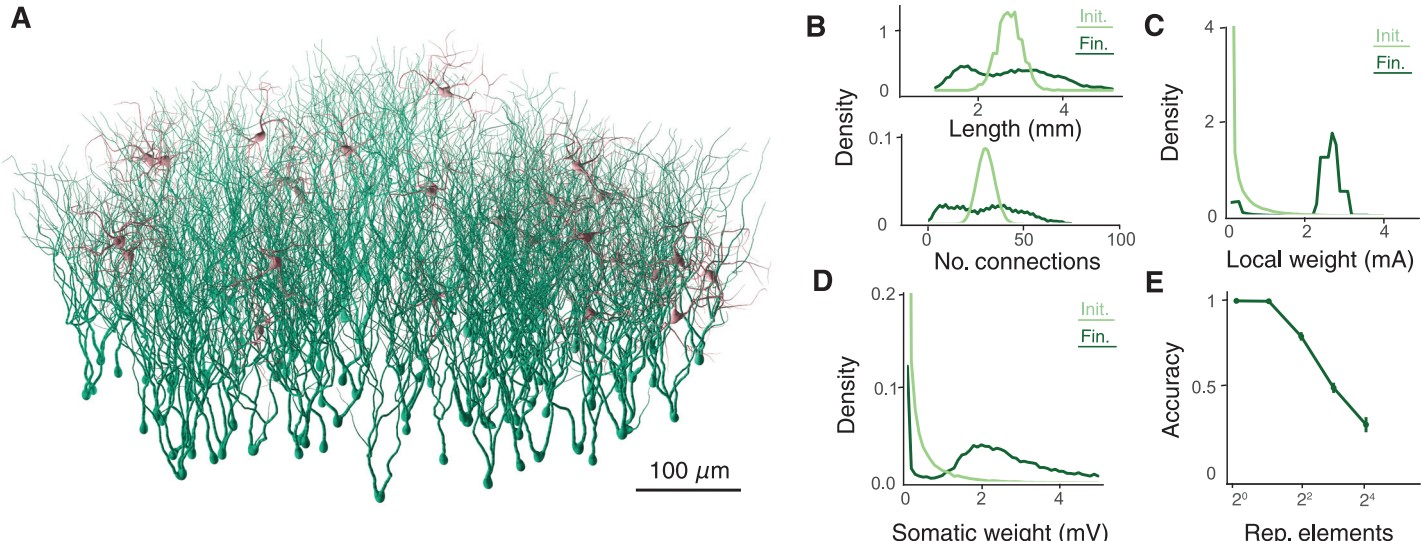

**Fig 5. Self-organisation in networks of spatially extended spiking neurons. A**, Spatially extended neurons with self-organised recurrent connectivity. Green dendrites are excitatory neurons and red dendrites are inhibitory neurons. **B**, Top: Distributions of dendrite lengths before (light green) and after (dark green) 50 epochs of learning. Bottom: Distributions of number of afferent contacts before (light green) and after (dark green) 50 epochs of learning. **C**, Distributions of local synaptic weights before (light green) and after (dark green) 50 epochs of learning. **D**, Distributions of somatic voltages induced by individual synapses before (light green) and after (dark green) 50 epochs of learning. All distributions are over stimuli with different numbers of repeated elements. **E**, Prediction performance of the spatially-extended neurons as a function of the number of repeated central elements. Error bars show ± one standard deviation over 5 repetitions.

an epoch, SET is applied to excise and replace the weakest excitatory-excitatory connections, and dendrites are resized to match their current level of innervation. After a number of epochs, synaptic plasticity is stopped and a standard artificial neural network is then trained to predict the next element in the input sequence from the instantaneous activity of the excitatory cells.

Fig 5 shows the performance of this network. The lengths of the excitatory dendritic trees change to reflect their afferent connectivity, a broader distribution of connectivities and sizes emerging Fig 5B). Local synaptic weights initially follow the specified gamma distribution (Fig 5C, light green) and are independent of the size of the neuron they impinge upon. After training, the local weights form a peaked distribution with reduced variability (Fig 5C, dark green). The combination of local weight and postsynaptic input size can be seen in the distribution of somatic voltages induced by each synaptic contact (Fig 5D), tending to be broader than the local distributions in both the untrained, but still dendritically normalised, and trained networks.

In terms of performance, the activity of the trained network forms a pattern that can be used to predict the next input in the sequence. The difficulty of this task depends upon the number of repeated elements in the input sequences (Fig 5E), but the spatially extended network is able to learn this task well for a range of different sequence lengths. In contrast the point neuron equivalent did not converge to a stable pattern of activity (see Methods). The original SORN network [49] employed an explicit $L^1$-normalisation alongside homeostatic spike-threshold adjustment [26] to maintain stable activity in the network; the implicit normalisation arising here from the sizes of the dendritic trees achieves a qualitatively similar result in allowing the network to converge to a stable state.

## Discussion

We have shown that the fact that excitatory synapses are typically located on dendrites produces an $L^0$ normalisation of synaptic inputs that improves the learning performance of sparse artificial neural networks with a variety of different structures. Such dendritic normalisation constrains the weights and expected inputs to be within relatively tight bands, potentially making better use of available neuronal resources. Neurons respond more to the proportion of their inputs that are active rather than the absolute number and highly-connected neurons are relatively less excitable. We believe that such a normalisation procedure is robust and should be applied to improve the performance of feedforward sparse networks.

Other results on normalisation [36, 37] have also demonstrated improvements in training performance in fully-connected feedforward networks. Such approaches work by keeping neurons relatively sensitive to changes in inputs and our results here can be seen as the sparse, and biologically justified, analogue, with similarly simple and broad applicability to the $L^2$-normalisation introduced by Salimans & Kingma (2016) [37]. In situations of dynamic connectivity, the dendritic normalisation outperforms other techniques. The comparison between the heterosynaptic plasticity-like $L^2$-normalisation and our dendritic $L^0$-normalisation is particularly interesting. In real neurons the former relies on actively re-weighting afferent contacts [25, 38] whereas the latter can arise purely from the passive properties of dendritic trees, and indeed would typically need the expenditure of additional energy to counteract. Neurons often display complementary functionality between passive structure and active processes, for example in the equalisation of somatic responses to synaptic inputs at different locations both dendritic diameter taper [50] and active signal enhancement [51, 52] play a role. Synaptic normalisation is in a similar vein. The effects are, however, distinct in some ways: while both normalisations keep neurons sensitive to inputs, the responses to learning differ. Heterosynaptic plasticity

enhances changes in the relative weighting of contacts, whereas dendritic normalisation increases the stability of well-connected neurons while allowing faster learning in poorly connected cells (Eq 5). This makes dendritic normalisation particularly suited to situations with evolving connectivity.

In the context of biological realism, the normalisation here has much room for development. We sought a straightforwardly demonstrable and quantifiable computational role for dendritic normalisation and so focussed on the most well-developed theories within artificial neural networks [53]. Such networks have a number of features that are impossible to implement in the brain, so more investigation into the benefits of size-invariant excitability in living systems is necessary. Firstly, a single artificial neuron can form connections that both excite and inhibit efferent cells, a phenomenon which is not seen in the connectivity of real neurons [54]. It is possible to regard the mix of excitatory and inhibitory connections as a functional abstraction of real connections mediated by intermediate inhibitory interneurons [55], but a more satisfying picture may emerge by considering distinct inhibitory populations of cells as we did in our self-organised network. Accordingly, it would be interesting to test the effects of implementing two distinct experimentally observed types of compartmental inhibition including somatic and dendritic inhibitory connections [56, 57]. Secondly, we mostly train our networks using supervised backpropagation which employs global information typically unavailable to real synaptic connections. Whilst there is emerging evidence that biological networks are able to approximate the backpropagation algorithm in some circumstances [9, 58, 59], a variety of other learning algorithms are also regarded as biologically plausible models for training networks [6, 7, 12, 16, 17]. Such algorithms are another natural fit for our normalisation as it too is implemented biophysically through the morphology of the dendritic tree. Thirdly, the neurons here generally are rate-based with either saturating or non-saturating outputs. Spiking networks can have different properties [11] and spikes could be incorporated into any of the approaches described above. The final sections of the results, with sparse recurrent and self-organising networks, address some of these issues by showing the contribution of dendritic normalisation to different types of learning and showing its generality as a computational principle, but there are many more interesting avenues to explore.

Dendrites in general have much more to offer in terms of artificial neural computations. Synaptic connections are distributed spatially over branched dendritic trees, allowing for a number of putative computational operations to occur within a single cell [60]. Dendrites are able to passively amplify signals selectively based on their timing and direction [18] and actively perform hierarchical computations over branches [61, 62]. Cuntz et al (2019) [23] noted that while mean neuronal firing rates are size-independent, the timing of individual spikes is not necessarily unaffected by morphology [22]. This means that signals encoded by rates are normalised whereas those encoded by spike timing may not be, implying that the two streams of information across the same circuit pathways are differentially affected by changing connectivity. Dendrites additionally hold continuous variables through their membrane potential and conductances that shape ongoing signal integration [63]. Such properties have potential computational roles that, while sometimes intuitive, have yet to be systematically studied at the level of neuronal circuits.

Overall, in line with the spirit of the emerging fruitful interaction between artificial intelligence and neuroscience [53, 64], this study has two major consequences. The first is a practical normalisation procedure that significantly improves learning in sparse artificial neural networks trained using backpropagation. The procedure can be used effectively for any sparsely-connected layers in shallow or deep networks of any architecture. Given that sparse layers can display better scalability than fully-connected layers [29], we believe that this procedure could

become standard in deep learning. Furthermore, the biological plausibility of the procedure means that it is highly appropriate as a component of more physiologically realistic learning rules. Secondly, we have taken the insights from Cuntz et al (2019) [23] and demonstrated a previously unappreciated way that the structure of dendrites, in particular their properties as leaky core conductors receiving distributed inputs, contributes to the computational function of neuronal circuits.

## Methods

### Cable theory

Eq 1 for the transfer resistance of a homogenous cable is derived in [23], following standard results in [18]. For a dendrite with an electrotonically compact soma of radius $\rho$, the soma has a leak conductance of $G_s(\rho) = 4\pi\rho^2 g_l$. In this case the transfer resistance (Eq 1) becomes

$$R_L(X) = \frac{\cosh(L - X)}{G_\infty(\sinh(L) + G_s(\rho)\cosh(L))} \tag{10}$$

and the moments of the somatic voltage change in response to synaptic plasticity at a random location are

$$\mu_{\Delta_v} = \frac{\tanh(L)\,\Delta_{syn}}{L\,G_\infty(\tanh(L) + G_s(\rho))} \quad , \quad \sigma^2_{\Delta_v} = \mu^2_{\Delta_v}\left(L^2\coth(L) + L\,\mathrm{cosech}^2(L) - 1\right) \tag{11}$$

The deviation from perfect inverse proportionality therefore depends on the ratio between $\tanh(L)$ and $G_s(\rho)$. For positive $L$, tanh grows monotonically between 0 and 1 and given typical values of $\rho = 5$, 10, and 15 $\mu$m, the steady state transfer resistances are plotted in Fig 1B (top panel).

For transient inputs, it is also necessary to consider the membrane specific capacitance $c$ and time constant $\tau_l = c/g_l$, giving a somatic impulse response at normalised delay $T = t/\tau_l$ of

$$\varepsilon(X, T) = \frac{e^{-T}}{L}\left[\frac{1}{2} + \sum_{n=1}^{\infty}\cos\left(\frac{n\pi X}{L}\right)e^{-\left(\frac{\pi n}{L}\right)^2 T}\right] \tag{12}$$

Taking expectations over synaptic location $X$ gives

$$\langle\varepsilon(T)\rangle = \frac{e^{-T}}{2L} \quad , \quad \langle\varepsilon^2(T)\rangle = \frac{e^{-2T}}{4L^2}\left[1 + \sum_{n=1}^{\infty}e^{-2\left(\frac{\pi n}{L}\right)^2 T}\right] \tag{13}$$

If a synaptic current $\zeta(t)$ takes the form of a difference of exponentials $\zeta(t) = \Theta_{syn}(e^{-t/\tau_f} - e^{-t/\tau_s})/(\tau_f - \tau_s)$ for fast and slow synaptic time constants $\tau_f$ and $\tau_s$ respectively, then the expected somatic voltage $\bar{v}(t)$ will be a convolution in time of $\zeta$ with Eq 13:

$$\bar{v}(t) = \frac{\Theta_{syn}\tau_l}{2L(\tau_f - \tau_s)}\left[\frac{\tau_f}{(\tau_f - \tau_l)}\left(e^{-t/\tau_f} - e^{-T}\right) - \frac{\tau_s}{(\tau_s - \tau_l)}\left(e^{-t/\tau_s} - e^{-T}\right)\right] \tag{14}$$

Similarly, the variance comes to

$$\mathrm{var}(v(t)) = \frac{\Theta^2_{syn}}{4L^2(\tau_f - \tau_s)^2}\sum_{n=1}^{\infty}\left[\frac{\tau_f\tau_n}{(\tau_f - \tau_n)}\left(e^{-t/\tau_f} - e^{-t/\tau_n}\right) - \frac{\tau_s\tau_n}{(\tau_s - \tau_n)}\left(e^{-t/\tau_s} - e^{-t/\tau_n}\right)\right]^2 \tag{15}$$

where $\tau_n = \tau_l\left(1 + \left(\frac{\pi n}{L}\right)^2\right)^{-1}$ for $n = 1, 2, \ldots, \infty$. Note that the mean voltage has an inverse

relationship with dendritic length independently of time. The expected influence of a change in synaptic strength on the response of the postsynaptic cell can then be quantified as either the peak voltage $\bar{v}_{max}$, or the total voltage received from a single synaptic activation $\bar{V} = \int_0^\infty \bar{v}(t)\,dt$. $\bar{v}_{max}$ does not have a compact analytical form, being $\bar{v}(t^*)$ where $t^*$ satisfies

$$(\tau_s - \tau_l)(\tau_f e^{-t^*/\tau_l} - \tau_l e^{-t^*/\tau_f}) = (\tau_f - \tau_l)(\tau_s e^{-t^*/\tau_l} - \tau_l e^{-t^*/\tau_s}) \tag{16}$$

but is plotted in Fig 1B (middle panel). $\bar{V}$ can be expressed simply as

$$\bar{V} = \frac{\Theta_{syn}\,\tau_l}{2\,L} \tag{17}$$

with variance

$$\text{var}(V) = \frac{\Theta_{syn}^2}{8\,L^2\,(\tau_f - \tau_s)^2} \sum_{n=1}^\infty \left[ \frac{\tau_f^2\,\tau_n^2}{(\tau_f + \tau_n)} + \frac{\tau_s^2\,\tau_n^2}{(\tau_s + \tau_n)} \right.$$
$$\left. - \frac{2\,\tau_f\,\tau_s\,\tau_n^2}{(\tau_f - \tau_n)(\tau_s - \tau_n)}\left( \frac{\tau_f\,\tau_s}{\tau_f + \tau_s} - \frac{\tau_f\,\tau_n}{\tau_f + \tau_n} - \frac{\tau_s\,\tau_n}{\tau_s + \tau_n} + \frac{\tau_n}{2} \right) \right] \tag{18}$$

and is plotted in Fig 1B (bottom panel).

## Network architectures

We first consider a simple artificial neural network (ANN) with one hidden layer to demonstrate the utility of our approach. The size of the input layer for both the MNIST and MNIST-Fashion datasets is 784, as each image is a $28 \times 28$ pixel greyscale picture. The hidden layer consists of $M$ neurons, each neuron $i$ receiving a number $n_i$ contacts from the previous layer. In Fig 1, $M = 30$, 100, and 300. In Figs 2 and 3, $M = 100$. Neuronal activation in the input and hidden layers as a function of input $z_i$ is controlled by a sigmoid function $\sigma(z_i)$

$$\sigma(z_i) = \frac{1}{1 + e^{-z_i}} \tag{19}$$

where $z_i$ is the weighted input to neuron $i$, given by

$$z_i = b_i + \sum_{n_i} w_{k,i} a_k \tag{20}$$

Here $b_i$ is the bias of each neuron $i$, $w_{k,i}$ is the synaptic weight from neuron $k$ in the previous layer to neuron $i$, and $a_k = \sigma(z_k)$ is the activation of presynaptic neuron $k$. The set of all $w_{k,i}$ for a given postsynaptic neuron $i$ form an afferent weight vector $\mathbf{w}_i$.

Both datasets have ten classes and the output of the ANN is a probability distribution assigning confidence to each possible classification. Neurons in the output layer are represented by softmax neurons where the activation function $\sigma_s(z_i)$ is given by

$$\sigma_s(z_i) = \frac{e^{z_i}}{\sum_{i=1}^{10} e^{z_i}} \tag{21}$$

The cost function $C$ is taken to be the log-likelihood

$$C = -\log(a_{\text{Correct}}) \tag{22}$$

where $a_{\text{Correct}}$ is the activation of the output neuron corresponding to the correct input.

**Table 2. Table summarising symbols and interpretations.** ANN symbols are defined above and cable theoretical symbols below the central line.

| Symbol | Interpretation |
|---|---|
| $a_i$ | Activation of neuron $i$ (Eqs 19 and 21) |
| $b_i$ | Bias of neuron $i$ (Eq 20) |
| $C$ | Cost function (Eqs 22 and 24) |
| $g_i$ | Excitability of neuron $i$ (Eqs 6 and 9) |
| $n_i$ | Number of afferent contacts to neuron $i$ (also written $\|\mathbf{v}_i\|_0$) |
| $s$ | (Uniform) Excitability of all neurons (Eqs 4 and 8) |
| $\mathbf{v}_i$ | Unnormalised input to neuron $i$ (Eqs 4 and 6) |
| $\mathbf{w}_i$ | Normalised input to neuron $i$ (Eqs 4 and 6) |
| $\varepsilon$ | SET connection probability |
| $\zeta$ | SET excision probability |
| $\eta$ | Learning rate for stochastic gradient descent |
| $\sigma$ | Sigmoid activation function (Eq 19) |
| $\sigma_s$ | Softmax activation function (Eq 21) |
| $\tau$ | Threshold-linear activation function (Eq 23) |
| $c$ | Membrane specific capacitance ($\mu$F/cm$^2$) |
| $g_l$ | Membrane leak conductivity (S/cm$^2$) |
| $G_\infty$ | Semi-infinite dendrite conductance ($S$, $\frac{\pi r^2}{\lambda r_a}$) |
| $l$ | Physical length of a dendrite ($\mu$m) |
| $L$ | Electrotonic length of a dendrite ($l/\lambda$) |
| $r$ | Dendrite radius ($\mu$m) |
| $r_a$ | Dendrite axial resistivity ($\Omega$cm) |
| $t$ | Time (ms) |
| $T$ | Normalised time ($t/\tau_l$) |
| $x$ | Physical distance along a dendrite ($\mu$m) |
| $X$ | Electrotonic distance along a dendrite ($l/\lambda$) |
| $\lambda$ | Electrotonic length constant ($\sqrt{r/2 r_a g_l}$, $\mu$m) |
| $\rho$ | Physical soma radius ($\mu$m) |
| $\tau_f$ | Fast synaptic time constant (s) |
| $\tau_l$ | Membrane leak time constant($c/g_l$, s) |
| $\tau_n$ | Effective time constant of $n$-th voltage mode ($\tau_l(1 + (\pi n/L)^2)^{-1}$, s) |
| $\tau_s$ | Fast synaptic time constant (s) |

For Fig 3, we generalise our results to deeper architectures and threshold-linear neuronal activations. In Fig 3A and 3B we expand the above to include 2 and 3 sparse hidden hidden layers, each with $M = 100$ sigmoid neurons. In Fig 3C and 3D we consider a simple convolutional neural network [46] with 20 $5 \times 5$ features and $2 \times 2$ maxpooling. In Fig 3E we return to the original architecture with $M = 100$, but replace the sigmoid activation function $\sigma(z)$ for the hidden neurons with a non-saturating threshold-linear activation function $\tau(z)$ defined by

$$\tau(z_i) = \max(0, z) \tag{23}$$

In Table 2, we show the results of other architectures to match published performance benchmarks. We replicate the published architecture in each case: For the original MNIST dataset and CIFAR-10 datasets, Mocanu et al [29] used three sparsely-connected layers of 1000 neurons each and 4% of possible connections existing. Pieterse and Mocanu [30] used the same architecture for the COIL-100 dataset. For the Fashion-MNIST dataset, Pieterse and

Mocanu [30] used three sparsely-connected layers of 200 neurons each, with 20% of possible connections existing.

For Fig 4, we generalise our results to recurrently connected networks. Here two input neurons are connected to a layer of 50 hidden neurons, which also have sparse recurrent connections. Inputs are given sequentially. The output is a single neuron with sequential output. The cost function is the mean squared error

$$C = \frac{1}{m} \sum_{j=1}^{m} \left( a_{j,\text{Output}} - a_{j,\text{Correct}} \right)^2 \tag{24}$$

where $a_{j,\text{Output}}$ is the activation of the output neuron, $a_{j,\text{Correct}}$ is the correct output, and $j$ indexes the input sequence. To classify accuracy, the raw value of each $a_{j,\text{Output}}$ is rounded to the nearest integer and the trial is considered accurate if all $j = 1, 2, \ldots, m$ rounded values are correct.

In all cases traditional stochastic gradient descent [40, 65] is used with a minibatch size of 10 and a learning rate $\eta$ of 0.05.

## Datasets

The ANN is originally trained to classify $28 \times 28$ pixel greyscale images into one of ten classes. Two distinct datasets are initially used. The MNIST, introduced by Lecun et al [40], consists of handwritten digits which must be sorted into the classes 0 to 9 (Fig 1B, left). The MNIST-Fashion dataset was introduced by Xiao et al [41] as a direct alternative to the original MNIST and consists of images of clothing. The classes here are defined as T-shirt/top, trousers, pullover, dress, coat, sandal, shirt, sneaker, bag, and ankle boot (Fig 1B, right). Each dataset contains 60, 000 training images and 10, 000 test images. State-of-the-art classification accuracy for the original MNIST dataset is as high as 99.77% [66], which likely exceeds human-level performance due to ambiguity in some of the images. For the newer MNIST-Fashion dataset state-of-the art networks can achieve classification accuracies of 96%. Such performance is achieved with deep network architectures, which we do not reproduce here, rather showing an improvement in training between comparable, and comparatively simple, artificial neural networks.

In Table 1, we also analyse other datasets. CIFAR-10 [67] contains 50, 000 training images and 10, 000 test images to be divided into the classes airplane, automobile, bird, cat, deer, dog, frog, horse, ship, and truck. Each image is $32 \times 32$ pixels in three colour channels. The COIL-100 dataset [68], which contains 7, 200 images in total, consists of images of 100 objects rotated in various ways. Each image is $128 \times 128$ pixels in three colour channels. There is no existing training/test split, so we follow Pieterse and Mocanu [30] in randomly assigning 20% of the available images to the test set.

## Networks of spatially extended neurons

In Fig 5, a recurrent network of spatially extended neurons self-organises in response to sequences of inputs. The task and overall network architecture is inspired by the reservoir computing demonstration in [49]. A number of excitatory $N_e$ and inhibitory $N_i$ neurons are chosen. Neurons are created, edited, and have their inputs integrated using the Trees Toolbox package in Matlab [69]. Excitatory dendritic trees are created by placing 50 target nodes uniformly randomly in a conical region with height 350 $\mu$m and terminal radius 125 $\mu$m. The soma of the neuron is placed at the point of the cone, and the target nodes are connected using the generalised minimum spanning tree algorithm `MST_tree`, with a balancing

factor of 0.7 [69, 70]. Inhibitory dendritic trees are created by placing 75 target nodes uniformly randomly in a spherical region with radius 100 $\mu$m. The soma of the neuron is placed at the centre of the sphere, and the target nodes were connected using the generalised minimum spanning tree algorithm `MST_tree`, with a balancing factor of 0.2 [69, 70]. Trees are resampled to have segments of 5 $\mu$m using the function `resample_tree`. The resulting initial distributions of tree lengths are plotted in Fig 5B. In both cases, intrinsic properties are taken to be radius $r = 1$ $\mu$m, axial resistivity $r_a = 100$ $\Omega$cm, and membrane conductivity $g_l = 5 \times 10^{-5}$ S/cm$^2$.

Sparse connectivity is initialised so that 30% of all possible excitatory-excitatory, 70% of all possible excitatory-inhibitory, and 70% of all possible inhibitory-excitatory connections exist. Neurons are resized so that their relative length corresponds exactly to their relative afferent connectivity, both inhibitory and excitatory. The mean length $\bar{l}$ and mean afferent connectivity $\bar{n}$ of the excitatory and inhibitory populations are calculated separately. Neurons are resized until the ratio of their length to afferent connectivity is identical to the population average. If a neuron is within 1% of the target length it is simply rescaled with the function `rescale_tree`. Otherwise, if it is too long, the shortest terminal branch is pruned by one node. If it is too short, an additional target point is added to the growth region defined above (conical for excitatory and spherical for inhibitory cells) and connected to the tree using the `MST_tree` function. The locations of the afferent synapses are then selected uniformly randomly from amongst the resulting dendritic segments.

All synapses are current-based for simplicity, with the (local) absolute magnitudes of synapses $|w|$ drawn from a gamma distribution $f_\gamma$ with mean 0.2 and standard deviation 0.2, giving shape $\alpha$ and rate $\beta$ parameters of 0.2 and 1 respectively:

$$f_\gamma(|w|) = \frac{\beta^\alpha \, |w|^{\alpha-1} e^{-\beta|w|}}{\Gamma(\alpha)} \qquad (25)$$

where $\Gamma(z) = \int_0^\infty x^{z-1} e^{-x} dx$ is the gamma function. The use of a gamma distribution ensures that synaptic magnitudes have the correct sign. The distribution of somatic voltage changes induced by these synapses are plotted in Fig 5C and depend on both individual weights and the size of the neuron on which they are located.

Activity propagates over discrete timesteps. At each timestep, all input currents to a neuron are used to evaluate the steady-state somatic voltage using the `sse_tree` function. If this is above a threshold the neuron spikes and induces currents at its efferent synapses for the next timestep. Existing excitatory-excitatory synapses undergo spike-timing dependent plasticity [5], where connections are strengthened if a presynaptic spike occurs the timestep before a postsynaptic spike and weakened if a postsynaptic spike occurs before a presynaptic spike. Specifically, local weights are defined to be between 0 and an upper limit synmax. Synapses potentiate by the learning rate parameter $\zeta$ multiplied by their distance to synmax and depress by $\zeta$ multiplied by their distance to 0. After a number, typically 1000, of stimulus presentations, the SET algorithm is applied to excise weak excitatory-excitatory connections and randomly add new connections with weights drawn from Eq 25. Excitatory dendrites are then resized using the above algorithm to match their new afferent connectivity and all afferent connections are randomly relocated on the resultant dendrite.

The learning tasks consists of random presentations of two 'words'. Each word consists of three 'letters', each of which is the random activation of a non-overlapping set of 5 input neurons, so that in total 30 excitatory neurons receive external input. One word can be thought of as abb...bbc, the other xyy...yyz with a different number of presentations of the middle letter in each trial. The network self-organises to reflect and predict the input patterns and the

difficulty of the task increases with the number of repetitions of the middle letter of each word. A standard neural network with 30 hidden neurons is used to classify the outputs. As the words are ordered randomly, the performance of the classifier is necessarily limited. The normalised accuracy score is 1 if the accuracy is perfect within a word and at chance level between words as in [49].

The point neuron equivalent is initiated identically to the above, but all synaptic currents are assumed to impinge directly on a spatially compact soma so that there is no relationship between length and connectivity.

For visualisation, dendrites are plotted in a column of radius 500 $\mu$m. Excitatory somata are randomly placed in a layer of depth 50 $\mu$m at the bottom of this column. Inhibitory somata are randomly placed in a layer of depth 200 $\mu$m that begins 100 $\mu$m above the excitatory layer. This organisation is for graphical purposes only and does not impose additional connectivity constraints.

## Supporting information

**S1 Fig. Supplementary figure.** A supplementary figure showing the performance of dendritic normalisation at different levels of sparsity.
(PDF)

**S1 Code. Supplementary code.** Code in Python and Matlab necessary to reproduce the figures. Code for Figs 1–4 is written in Python 3.6. The networks in Figs 1, 2 and 4 are coded using the standard Numpy package, and the networks in Fig 3 make use of Keras with a TensorFlow backend (`keras.io`). The application of dendritic normalisation in Keras with TensorFlow allows for immediate inclusion in Keras-based deep learning models. The normalisation requires a custom layer, constraint, and optimiser. Fig 5 uses code written in Matlab 2020b, using the freely available Trees Toolbox package [69].
(ZIP)

## Acknowledgments

We would like to thank F Effenberger and K Shapcott for helpful discussion.

## Author Contributions

**Conceptualization:** Alex D. Bird, Hermann Cuntz.

**Data curation:** Alex D. Bird.

**Formal analysis:** Alex D. Bird, Peter Jedlicka, Hermann Cuntz.

**Funding acquisition:** Alex D. Bird, Peter Jedlicka, Hermann Cuntz.

**Investigation:** Alex D. Bird, Peter Jedlicka, Hermann Cuntz.

**Methodology:** Alex D. Bird, Peter Jedlicka, Hermann Cuntz.

**Project administration:** Alex D. Bird, Peter Jedlicka, Hermann Cuntz.

**Resources:** Alex D. Bird, Peter Jedlicka, Hermann Cuntz.

**Software:** Alex D. Bird, Hermann Cuntz.

**Supervision:** Alex D. Bird, Hermann Cuntz.

**Validation:** Alex D. Bird, Peter Jedlicka, Hermann Cuntz.

**Visualization:** Alex D. Bird, Peter Jedlicka, Hermann Cuntz.

**Writing – original draft:** Alex D. Bird.

**Writing – review & editing:** Alex D. Bird, Peter Jedlicka, Hermann Cuntz.

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
