## [Decision Letter · Decision Letter 0]

3 Nov 2020

Dear Dr Bird,

Thank you very much for submitting your manuscript "Dendritic normalisation improves learning in sparsely connected artificial neural networks" for consideration at PLOS Computational Biology.

As with all papers reviewed by the journal, your manuscript was reviewed by members of the editorial board and by several independent reviewers. In light of the reviews (below this email), we would like to invite the resubmission of a significantly-revised version that takes into account the reviewers' comments.

A most important point, as raised in the reviews, concerns the biological relevance of your results. This is essential for being considered for publication in PLoS CB; otherwise, the results may be more suited to a more specialized journal, e.g. reflecting the machine learning aspect of the work;

We cannot make any decision about publication until we have seen the revised manuscript and your response to the reviewers' comments. Your revised manuscript is also likely to be sent to reviewers for further evaluation.

Sincerely,

Lyle Graham

Deputy Editor

PLOS Computational Biology

Reviewer's Responses to Questions

**Comments to the Authors:**

Reviewer #1: The authors cite a biologicial motivation for the use of L0 weight normalization in sparse neural networks. They then show how this normalization improves performance over unnormalized networks on several different architectures and tasks. It’s good that the authors show results on many different architectures and in comparison to other normalizations and I also appreciate the comparison to models in the existing literature. However, I found the paper overall to be rather unclear both in the description of the work and its interpretation. Furthermore, its relevance to a biological audience could be greatly enhanced.

*In the Introduction it is unclear when findings relate to feedforward vs recurrent networks. For example, small-worldness (as I know it) is a property of recurrent networks.

*The SET algorithm needs to be explained better, and more so in the main text. The methods section says “After each training epoch, a fraction of the weakest contacts are excised and an equal number of new random connections are formed”. Is this specific to each neuron such that the number of connections per neuron doesn’t change or is it that the number of connections in the entire network is conserved? If it is the former then the normalization would simply be division by a (neuron-specific) constant, which I thought may be the case given the statement on line 94. But then Figure 2B shows contacts per neuron changing over training so I assume it is the latter. This is an important point to be clear on.

*I am a bit confused as to how the different normalizations compare and what the authors are claiming about their dendritic normalization.

For example: “The normalisation appears to make better use of neural resources by distributing connections and weights more evenly across the available cells, whilst keeping expected inputs closer to 0, the steepest part of the activation function, where responses are most sensitive to inputs”

First, does “whilst keeping expected inputs closer to 0” align with what’s plotted? The distribution for the normalized model in the right column of Figure 2B starts centered around 0 but spreads out with training such that is seems like most cells get input far from 0, similar to the unnormalized model. Also, is this the main explanation of the benefit of the L0 norm? How does this compare to the benefit of the L2 norm? That is, are the authors claiming that the L0 norm is having a qualitatively different effect than L2 that makes the L0 norm beneficial in its own way?

"In consequence, cells receiving a few strong connections will be relatively fast-learning and unstable compared to those that receive many, individually less effective, connections." Doesn’t the L2 norm mean that cells with strong connections will learn slowly? In fact, it seems like the L0 and L2 norms fight each other (especially if overall weight is conserved). It may be helpful to simply plot a measure of the gradients for the weights a cell receives as a function of the number of contacts a cell gets and/or its overall weight input, for the different norm types. That way the differences between L2, L0, and the combination can be clearly and empirically shown.

"As neurons with many weak connections learn relatively slowly when both normalisations are in place it appears that good use is made of existing connectivity, but that less informative connections are not selectively weakened enough to be excised by the SET algorithm." I find this statement a bit hard to parse. Is it just saying that the slow learning essentially means that the SET part of the learning process isn’t relevant? Also, why wouldn’t this be true for L0 alone?

"The joint mechanism appears well suited to sparse, but static, connectivity" It seems like the joint mechanism results in static connectivity (and worse performance). I don't know if that makes it well suited for that, per se.

*" The dendritic normalisation (L0-norm in pink) has the highest performance after 10 epochs, although the L1-norm with variable excitability is similar. " Why do the authors report on 10 epochs when training goes to 20? Is this statement true for 20 as well? Can you include in-line here the performance +/- SD and whether this is significantly different from the next best one. It's not so easy to see on the graph.

*"here the two streams of input would be in competition as more strongly recurrently connected cells would receive relatively weak feedforward inputs and vice versa." why would this be?

*Overall, I am struggling to find a clear demonstration of what is unique and relevant about the L0 norm. It seems to lead to relatively similar performance as L2 and aside from some brief suggestions about what the differences may be, there isn’t any actual analysis of how L0-normed networks learn differently than L2 ones (while it is interesting that the L0 norm has its own biological motivation, that is not enough.). If the L0 norm is just doing something that is mostly the same as any other norm (i.e., keeping weights within a reasonable range) then I’m not sure how interesting it is to the biological community. The authors even say that “The comparison between the heterosynaptic plasticity-like L2-normalisation and our dendritic L0-normalisation is particularly interesting.” I agree, but I don’t feel that this is actually done to a great extent in the paper, aside from some speculating. Given that this is a computational biology journal rather than a machine learning one, I think the authors need to put in more effort to provide some insights into how the system works, and also how this relates to how real neurons work. This is especially important if the main benefit of the normalization is how it controls learning rates, given that the learning algorithm (backprop) is not directly relatable to biological learning. That is, the authors should be able to clearly state what they believe the implications from this somewhat non-biological learning are for biological learning.

I should also point out that one of the main contributions (“a practical normalisation procedure that drastically improves learning in sparse artificial neural networks trained using backpropagation.”) seems to not hold when compared to existing L2 norms.

Minor Comments:

It’s difficult to see the shading in Figure 1d

Line 163: what is this sigma? I don't see it in the equations or previously referenced.

Reviewer #2: The authors investigate the benefits of normalizing synaptic weight strength by the number of postsynaptic connections in sparse neural networks with evolving connectivity. More concretely, the authors consider standard artificial feedforward neural network models trained with end-to-end stochastic gradient descent; an additional connectivity remodeling process ("SET") replaces weak connections by random new ones as training proceeds. The proposed normalization is a natural extension of weight normalization techniques to the sparse, non-homogeneous connectivity case. As the authors comment in the abstract, their research essentially results in "improved machine learning tools".

While I appreciate the simplicity of the method from a practical machine learning perspective, I feel that the paper is not very well matched to the PLOS Computational Biology readership. There is an interesting connection to previous biophysical modeling work from the same group, but I think it is fair to say that the normalization can be easily derived from purely theoretical considerations. I find it hard to call this work "bio-inspired machine learning". Apart from this connection, the research is essentially standard machine learning research.

Thus, I would recommend submitting this work to an artificial neural networks venue with less emphasis on biology than PLOS Computational Biology, or thoroughly rewriting the paper having a non-machine-learning readership in mind. Admittedly, it may be hard to do so.

I leave some additional comments to the authors below:

- I suppose that the authors are aware that as a machine learning paper, the experiments are lacking by today's standards in the choice of architectures and range of problems considered. It may be worth considering recurrent neural networks, given the difficulties of optimizing them and their relevance in computational neuroscience.

- The most surprising result, for me, was that constant excitability was beneficial (around l. 200). The paper would be substantially stronger if the authors provided more experimental evidence in favor of this finding, as the per-neuron excitability is usually featured in batch and weight normalization and believed to be beneficial to improve optimization in deeper and more complex neural network models.

- It would be good to know how the interplay between the spherical and L0 weight normalizations is affected when considering less sparse, but still SET-rewired networks. A related question, how do the hyperparameters of the SET algorithm, which were kept fixed here, affect the results?

- Why are the contact distributions bell-shaped (Fig. 2) and far from the more interesting "small-world and scale-free topologies similar to biological neuronal circuits (31)" (l. 59) referred to by the authors?

- The authors conclude the abstract by claiming that their method "renders [sparse networks] better understood and more stable in our tests", after stating that artificial neural networks "result in poorly understood parameters". After reading the paper I was puzzled by these claims, as I couldn't see why the proposed normalization is related to any improved parameter interpretability.

Reviewer #3: Summary:

Through simulations of artificial neural networks, this work makes explicit that having dendrites constraining synapses to scale their weight with the inverse of their number (i.e. normalization) comes with foreseeable advantages for learning, when learning synapses learn according to the backpropagation-of-error algorithm. The advantages are mainly in allowing for a faster progression of learning and are robust to the type of normalization, the number of neurons in a layer and the depth of the network.

Assessment:

I found myself oscillating between the excitement at seeing a clear function of all dendrites for learning and the deception of the fact that the paper is a small adaptation recast with an introduction on dendrites that weight normalization may be beneficial for learning. I think that a revised presentation can mitigate the deception.

Some important criticisms:

1. Rationale. The paper is evasive about the rationale and the result section has none. From what I can see, the simulations are exactly as in some other ML papers, with the exception of combining normalisation and sparsity. The reader appreciates that these papers are presented form the get-go, but we are missing a rationale for delving into the variants studied here. Why do we need to focus on sparse nets? Do we expect anything different or is this a simple sanity check for this more biologically realistic constraint? If it is studied for biological realism, why any of the other issues with biological realism, such as those mentioned in the discussion. There is more rationale given in the discussion (li 275, etc) than in the results section.

2. Gradient-based learning. The fact that these results are tied to training with backprop should be ideally part of the rationale. The results presented here would hold only if the neurons are actually learning with backprop. There is work in this direction (see Sacramento et al. 2018, Payeur et al. 2020, Bellec et al. 2020; and these papers are clearly making the claim that it is backprop can be realistic, so it is (thankfully for the present paper) not true that the gradient information is necessarily unavailable to synapses as mentioned in the discussion and introduction).

3. Validity of the premises. The entire paper being based on the idea that dendrites implement a normalization. Now for this fact, the authors reference their prior work on bioRxiv and the reader is supposed to just take this for granted. But having read the reference, I find it less than obvious that this is necessarily the case, or that the referenced paper actually serves a good reference for this fact since it is not the main point of that paper. The reference does not have an explicit figure on normalisation, something along the line of effective weight against number of synapses. Or if there is one, I did not get it. We therefore do not know the level with which this is an accurate picture of the biophysical properties of dendrites, we do not know where this approximation breaks down nor if it does at all. Further, a more solid establishment of this paper in the biological--as opposed to the machine learning--realm is specifically what is lacking from the narrative. In thinking about this, I am not able to see why this normalization property has be down to dendrites only. So there is an inference step that the authors are making in establishing the premises of the study that is not obvious. I find the absence of a specific validation of the premises concerning and I am sure that other readers will have the same worry. (I would even go so far as to say that the claim in the introduction that ‘we show how the dendritic morphology of a neuron produces an afferent weight normalisation’ is blatantly wrong).

Minor points

1. I did not understand the content of the paper from the abstract. For instance, there is no mention of dendrites. Learning comes out of the blue. The paper is said to be about introducing a normalisation but that would be missing the point.

2. The introduction does not situate this work as being part of a number of other studies in finding a specific network-level function of dendrites. Instead, the introduction focused more on the details that make ANNs slightly better.

3.

4. Ambiguous statement on the nature of the normalisation. The abstract implies that it is normalised by ‘the number of active inputs’ and Eq. 1 states something similar with the L0 norm (but I am confused as vi is not defined, is it activity or some type of raw weight?). But then much of the text is about the sheer number of connections.

5. The vector vi is not defined.

6. The cost function C is not defined.

7. Eq. 2 seems wrong. Perhaps an explanatory step is needed: the gradient over v should give two terms when applied to equation 1 because v_i appears in both in itself and in the normalisation.

8. Li 105 states that the networks are trained with SET, but that is not entirely true: missing that they are trained with SET and SGD with backprop.

9. The ‘performance’ or the statements that ‘learning is improved’ is not clearly defined and seem to oscillate between meaning accuracy and learning speed. Please make more precise throughout.

10. The comparison on accuracies is not very convincing between normalized and unnormalized as many networks have not finished learning, muddling learning rate and accuracy together.

11. The point about the greater reliability is not illustrated clearly, the difference in error bars is nor consistent nor obvious (it seemed to be violated in Fig. 3).

12. More explanation as to why we expect L2 norm to be implemented by heterosynaptic mechanisms is needed. As it stands, it is as though these two processes are synonymous.

13. Li 146 ‘the improvement seems to increase with complexity’. This vague statement should either be established firmly (I mean the simulations are there) or removed.

14. Li 199 ‘All normalisation show substantial improvement over the control case (Fig 1d)’ I did not understand what Fig 1d had to do with this, probably meant 3g, but then that is not what 3g shows either.

15. Fig. 3g seems out of place in figure 3 since it is about something entirely different than depth (the title of figure 3) and the idea of looking at different types of normalization would need to be fleshed out.

16. Li 290 ‘drastically improves’ is an overstatement.

**Have all data underlying the figures and results presented in the manuscript been provided?**

Reviewer #1: None

Reviewer #2: None

Reviewer #3: **No: **See major point 3 in the comments to authors.

PLOS authors have the option to publish the peer review history of their article (what does this mean?). If published, this will include your full peer review and any attached files.

Reviewer #1: No

Reviewer #2: No

Reviewer #3: No
---

## [Decision Letter · Decision Letter 1]

25 Mar 2021

Dear Dr Bird,

Thank you very much for submitting your manuscript "Dendritic normalisation improves learning in sparsely connected artificial neural networks" for consideration at PLOS Computational Biology. As with all papers reviewed by the journal, your manuscript was reviewed by members of the editorial board and by several independent reviewers. The reviewers appreciated the attention to an important topic, but also identified several points that should be addressed. Based on the reviews, we are likely to accept this manuscript for publication, providing that you modify the manuscript according to the review recommendations.

In addition, can you verify that S2 Code and Data has been uploaded, so that the reviewers may access this?

Sincerely,

Lyle Graham

Deputy Editor

PLOS Computational Biology

[LINK]

Reviewer's Responses to Questions

**Comments to the Authors:**

Reviewer #1: I appreciated the added effort the authors have put into the work, particularly the recurrent architectures and spiking network. However there are still some areas that need corrections/clarification.

This statement is added to the Introduction and it is not clear to me what part of the results it refers to. Is it just trying to say that the experiments use backprop? It should be made more clear: "secondly we demonstrate that learning is improved by

dendrites in the ideal case where all gradient information is available to every synapse."

This statement (now in the Results section) has still not been clarified to say that this is happening over the whole network, rather than individual neurons:

"After each training epoch, a fraction ζ of the weakest contacts are

excised and an equal number of new random connections are formed. New connection

weights are distributed normally with mean 0 and standard deviation 1."

Furthermore, if the number of connections per neuron is changing why do the authors say it is almost everywhere constant in line 132?

The authors highlight when discussing the results of Figure 1 that the benefits to learning from normalization can be seen not just in test accuracy, but also in training cost and in variability over random initializations. Only one of these (benefits as measured on test accuracy) holds in Figure 3, yet the authors say "In all cases, learning is improved by dendritic normalisation (orange

versus blue lines in Figures 3b and d for the MNIST-Fashion dataset) in a similar

manner to that for the single-layered networks described above." This is simply not true and needs to be rewritten in a way that accurately describes the figure.

The claim on lines 390-393 needs to be made more explicit, i.e. say what the performance of that algorithm is compared to this one.

line 319: "network network" typo

Reviewer #3: "Dendritic normalisation improves learning in sparsely connected artificial neural networks" was quite seriously and adequately revised. The authors addressed all of my concerns. I find the article reads much better and its interest for the computational neuroscience is clear and convincing. The added derivation at the beginning is excellent and fits very well. The addition of BPTT makes it quite impressive. I commend the authors for the careful and exhaustive revision.

Two minor comments:

In the (new) first 2 paragraphs of the results section, it should be stated early on that the derivation is for the stationary state. Along those lines, it can be nice for the reader that the mean and variance (Eq 2) are over spatial configurations of the synapses.

The performance of 90% (reported in the figures) for MNIST is on the low side for a 3 layer network. But then the table speaks of an accuracy of 99% in the table. It would be nice to have a clear explanation of the mismatch between the accuracy in the table and in the figure.

I still think that the field as changed such that the phrase 'local algorithms are still regarded as a more plausible model for training networks' does not holds, particularly when used in opposition to some of the biological approximation to backprop. All of the biological approximations to backprop are meant to give a local algorithm. Perhaps the authors meant something closer to the ML 'unsupervised'?

Richard Naud

**Have all data underlying the figures and results presented in the manuscript been provided?**

Reviewer #1: None

Reviewer #3: Yes

PLOS authors have the option to publish the peer review history of their article (what does this mean?). If published, this will include your full peer review and any attached files.

Reviewer #1: No

Reviewer #3: **Yes: **Richard Naud

Figure Files:

Data Requirements:

Reproducibility:

References:

---

## [Decision Letter · Decision Letter 2]

19 Jun 2021

Dear Dr Bird,

We are pleased to inform you that your manuscript 'Dendritic normalisation improves learning in sparsely connected artificial neural networks' has been provisionally accepted for publication in PLOS Computational Biology.

Best regards,

Lyle J. Graham

Deputy Editor

PLOS Computational Biology

Reviewer's Responses to Questions

**Comments to the Authors:**

Reviewer #1: None

**Have the authors made all data and (if applicable) computational code underlying the findings in their manuscript fully available?**

Reviewer #1: Yes

PLOS authors have the option to publish the peer review history of their article (what does this mean?). If published, this will include your full peer review and any attached files.

Reviewer #1: No

---

## [Editor Report · Acceptance letter]

15 Jul 2021

PCOMPBIOL-D-20-01654R2 

Dendritic normalisation improves learning in sparsely connected artificial neural networks

Dear Dr Bird,

I am pleased to inform you that your manuscript has been formally accepted for publication in PLOS Computational Biology. Your manuscript is now with our production department and you will be notified of the publication date in due course.

With kind regards,

Katalin Szabo
